



# Measurement Report: An investigation of the spatiotemporal variability of aerosol in the mountainous terrain of the Upper Colorado River Basin from SAIL-Net

Leah D. Gibson[1,a], Ezra J.T. Levin[1,b], Ethan Emerson[1,b], Nick Good[2,c], Anna Hodshire[1,b], Gavin McMeeking[1,c], Kate Patterson[1,a], Bryan Rainwater[1,b], Tom Ramin[1], and Ben Swanson[1,d]

[1]Handix Scientific, Fort Collins, CO, USA
[2]Good Science, Fort Collins, CO, USA
[a]now at: Colorado Department of Public Health and Environment, Denver, CO, USA
[b]now at: METEC Research Group, Colorado State University Energy Institute, Fort Collins, CO, USA
[c]now at: CloudSci, Fort Collins, CO
[d]now at: College of Civil and Environmental Engineering, Colorado State University, Fort Collins, CO, USA

**Correspondence:** Leah D. Gibson (lgibson@handixscientific.com)

**Abstract.** In the Western US and similar topographic regions across the world, precipitation in the mountains is crucial to the local and downstream freshwater supply. Atmospheric aerosols can impact clouds and precipitation by acting as cloud condensation nuclei (CCN) and ice nucleating particles (INP). Previous studies suggest there is increased aerosol variability in these regions due to the complex terrain, but none have quantified the extent of this variability. In fall 2021, Handix Scientific

contributed to the US Department of Energy (DOE)-funded Surface Atmosphere Integrated field Laboratory (SAIL) in the East River Watershed (ERW), CO, USA by deploying SAIL-Net, a novel network of six aerosol measurement nodes spanning the horizontal and vertical domains of SAIL. The ERW is a topographically diverse region where single measurement sites can miss important observations of aerosol-cloud interactions. Each measurement node included a small particle counter (POPS); a miniature CCN counter (CloudPuck); and a filter sampler (TRAPS) for INP analysis. SAIL-Net studied the spatiotemporal

variability of aerosols and the usefulness of dense measurement networks in complex terrain. After the project's completion in summer 2023, we analyzed the data to explore these topics. We found increased variability compared to a similar study over flat land. This variability was correlated with the elevation of the sites, and the extent of the variability changed seasonally. These data and analysis stand as a valuable resource for continued research into the role of aerosols in the hydrologic cycle and as the foundation for the design of measurement networks in complex terrain.

## 1 Introduction

In mountainous regions, winter snowpack and overall precipitation are vital for maintaining local and downstream freshwater supplies. In these areas, atmospheric aerosols play a role in local precipitation patterns, acting as cloud condensation nuclei (CCN) and ice nucleating particles (INP) (Jirak and Cotton, 2006; Lynn et al., 2007). It is therefore critical to understand and monitor aerosol concentrations in these areas. Ambient aerosols are spatially and temporally complex due to their many sources



and relatively short atmospheric lifetimes (Anderson et al., 2003; Weigum et al., 2016). This complexity is further amplified in mountainous terrain (Zieger et al., 2012; Yuan et al., 2020; Nakata et al., 2021). Direct measurements of aerosols across spatial and temporal scales are therefore essential to fully understand the role of aerosols in cloud formation and precipitation.

Orographic clouds created by topographically forced upward motion are an important contributor to winter snow in mountainous regions. In these clouds, ice crystals form in the upper layers and then fall through a supercooled liquid layer, collecting

rime and growing larger before reaching the ground as snow or graupel. This process is sensitive to the amount of CNN and INP present (Creamean et al., 2013; Levin et al., 2019). The amount of riming gathered by descending crystals is contingent upon the size of supercooled liquid droplets, where smaller droplets are less efficiently collected. In CCN-rich clouds, droplets are smaller, resulting in reduced rime and overall precipitation. In the Rocky Mountains of Colorado, Saleeby et al. (2011) found that decreased riming causes a shift in precipitations from windward to leeward slopes and potentially into different

watersheds. The riming process is also oppositely influenced by INP, where higher concentrations of INP increase precipitation (Rosenfeld et al., 2014). Thus, understanding the spatial and temporal variability of atmospheric aerosols is necessary to understand the role of aerosols in the hydrologic cycle in mountainous regions and the subsequent impacts on freshwater availability.

To further study land-atmosphere interactions and their impact on the hydrologic cycle in mountainous regions, the US De-

partment of Energy (DOE) supported the Surface Atmosphere Integrated field Laboratory (SAIL) in the East River Watershed (ERW) of the Upper Colorado River Basin in southwestern Colorado. The Colorado River Basin covers parts of Colorado, Utah, Nevada, New Mexico, California, and all of Arizona. These states withdraw an average of 17 million acre-feet of water each year (Maupin et al., 2018). In the past 20 years, the basin has experienced increasingly intense droughts, leading to concern over freshwater availability in the Western United States. Precipitation is affected by anthropogenic aerosols, and it is

estimated that the Colorado River Basin loses approximately 538,0000 acre-feet of water each year due to an increase in CCN caused by anthropogenic emissions (Jha et al., 2021). Thus, one of the main goals of the SAIL campaign was to improve earth system modeling to better predict the timing and availability of water resources from the mountains in this region.

Two monitoring sites were deployed in the East River Watershed from fall 2021 to spring 2023 as part of SAIL. The two sites were the Aerosol Observation System (AOS) located on Crested Butte Ski Mountain, and the ARM Mobile Facility

(AMF-2), located at the Rocky Mountain Biological Laboratory in Gothic, Colorado. Both sites collected a variety of aerosol and atmospheric measurements (Feldman et al., 2023). While these two sites provided comprehensive aerosol measurements, they may not have fully represented the complete spatial variability of aerosol concentrations due to the complex terrain of the region (Schutgens et al., 2017). Thus, additional measurement locations were beneficial, if not crucial, to understanding aerosol-cloud interactions in complex terrain.

To gain a more comprehensive understanding of aerosols in the region, we deployed SAIL-Net, a distributed network of six measurement nodes spanning the domain of the SAIL research area from October 2021 to July 2023. Each node measured aerosol particles between 140 nm and 3.4 $\mu$m in diameter using a small particle counter (POPS, (Gao et al., 2016)), CNN using a miniature CCN counter (CloudPuck), and INP using the Time-Resolved Aerosol Filter Sampler (TRAPS, Creamean et al. (2018)). Our approach was similar to other studies that aimed to better characterize and understand aerosols and gas-phase



pollutants using networks of lower-cost sensors (Caubel et al., 2019; Kelly et al., 2021; Asher et al., 2022). Such studies have identified neighborhood-level variations in pollutant concentrations (Schneider et al., 2017; Popoola et al., 2018; Caubel et al., 2019). Small-scale variations such as this are poorly represented in models and poorly measured by a single monitoring system (Caubel et al., 2019). Previous work has shown the representation error (the ability of measurements to represent a larger area) increases with complex orography, leading to decreases in model accuracy (Schutgens et al., 2017). The overall goal of SAIL-

Net was to improve our understanding of the variability of aerosol in the ERW, thus increasing our knowledge of aerosol-cloud interactions in this region and informing the usefulness of distributed networks of measurements for future studies. We met this goal by answering the following science questions:

1.  **What is the aerosol temporal variability, and how does aerosol inhomogeneity vary seasonally?** Is there significant seasonal variability in sources, or are short-term meteorological conditions the most important determining factor in sources for cloud nuclei?


2.  **What is the aerosol spatial variability?** What are the aerosol characteristics at cloud base, presumably the particles most representative of those acting as cloud nuclei?

3.  **How should measurement networks be designed to capture aerosol-cloud interactions, and what do they need to measure?** Can a single measurement site accurately represent aerosol properties in regions of complex terrain?

The goal of this paper is to introduce SAIL-Net, highlight initial observations of the POPS data, and use these findings to address the science questions. We hope these data and analyses inspire future research in studying the variability and impact of aerosol in mountainous terrain.

Section 2 of this paper introduces the instrumentation, sites, and data of SAIL-Net. Next, Sect. 3 uses the data from the POPS to address our scientific questions and highlights the trends we have seen in the data. This is broken into three subsections. First,

Sect. 3.1 identifies the temporal variability of aerosol in the ERW by looking at seasonal and diurnal patterns. Next, Sect. 3.2 highlights the variability of aerosol in the region and suggests conditions and sources that may affect this variability. Lastly, in Sect. 3.3, we comment on the network as a whole to determine if a single measurement site could sufficiently represent the ERW.

## 2 Methods

Each site included a suite of three relatively low-cost, lightweight microphysics instruments manufactured by Handix Scientific to measure aerosol size distributions (POPS), CCN concentrations (CloudPuck), and INP concentrations (TRAPS). Together this network of instruments formed a comprehensive picture of aerosol-cloud interactions in the region. These instruments were chosen because their size, price, low power requirements, and self-sufficiency were the optimal combination to support a distributed network of sites in remote terrain.



The three instruments were secured inside a weatherproof enclosure and mounted on 10-foot tall scaffolding to keep the instruments above the snow in the winter. The inlets of the instruments faced downward and were protected by a baffle. Four of the six sites ran on solar power while the other two sites used established ground power sources.

This manuscript will focus on data from the Portable Optical Particle Spectrometer (POPS). The POPS is a small, low-cost optical particle counter initially developed at NOAA by Gao et al. (2016) and commercialized by Handix Scientific. In the last few years, it has been established as a research-grade instrument (Yu et al., 2017; Mei et al., 2020; Brus et al., 2021). The instrument measures the intensity of light scattered by particles passing through a 405 nm laser to optically size particles into user-selectable size bins between approximately 140 nm and 3.4 $\mu$m, and measures at a one-second resolution.

The POPS operated continuously at each SAIL-Net node, except during power outages, deep snows that buried some inlets, or other instrument malfunctions. This was the largest and longest dataset produced during SAIL-Net.

## 2.1 Network description

SAIL-Net consisted of six measurement nodes spread across the ERW near Crested Butte, CO. The primary objective in site placement was to select locations that captured the vertical variation in aerosol properties while also spanning the domain of the SAIL campaign. The elevation of the sites ranged from roughly 2750 m along the valley floor of the ERW to approximately 3500 m near the top of Crested Butte Mountain, which is one of the taller peaks in the ERW. The farthest distance between sites was 14 km, while the closest two sites were approximately 1 km apart. The disparate elevations of the sites resulted in different vegetation surrounding the sites. Table 1 describes each site. A map of the sites is also provided in Fig. 1, and Fig. 2 provides a photo of each site.

## 2.2 Data acquisition and post-correction

SAIL-Net sites were visited approximately monthly. During each visit, a suite of checks were performed to ensure instrument reliability and to document instrument drift. The POPS underwent the most checks and monitoring. We checked the inlet flow of the instrument and recorded the accuracy of the POPS in sizing 500 nm aerosolized polystyrene latex beads (PSL check). This information was used to later post-correct the data. We did not recalibrate the POPS in the field to correct drift at any point during the campaign in order to avoid causing discontinuities in the raw data. However, if any of the instruments required major repairs, the instrument was removed, repaired, or replaced, and returned the following month. When a new or repaired POPS was returned to the field, its sizing had been recalibrated. In these cases, there was some discontinuity in sizing accuracy, but these were corrected in the post-analysis data as discussed below.

The data correction process focused on correcting drift in the POPS sizing accuracy. All POPS instruments in SAIL-Net experienced some drift, but the drift rate and amount were not uniform across the different instruments. We collected data from PSL checks for the majority of site visits, but not all. Some sites were not visited during certain months due to accessibility issues, or the PSL check was not performed due to instrument malfunctions or weather. Thus, some assumptions were made during post-correcting to account for these gaps. We assumed that POPS were performing at their factory calibration level at the start of the measurement period in fall 2021 (or summer 2022 in the case of CBTop), and therefore did not need post-correction



| Site Name | Location | Elevation | Deployment Months | Description |
|---|---|---|---|---|
| Pumphouse | 38.9211°N, 106.9495°W | 2765 m | Oct. 2021-July 2023 | Instrumentation was mounted on scaffolding and ran on solar power. Located in a meadow in the East River Valley next to the East River. |
| Gothic | 38.9561°N, 106.9858°W | 2918 m | Oct. 2021-July 2023 | Colocated with AMF-2 in a meadow near Gothic, also in the East River Valley. Instrumentation was mounted on scaffolding and ran on ground power. Higher traffic and human activity nearby in the summer. |
| CBMid | 38.8983°N, 106.9431°W | 3137 m | Oct. 2021-June 2023 | Colocated with AOS on Crested Butte Ski Resourt. Instrumentation was mounted on AOS trailer and ran on ground power. Near a groomed ski run in the winter. |
| Irwin | 38.8874°N, 107.1087°W | 3177 m | Oct. 2021-July 2023 | Instrumentation was mounted on scaffolding and ran on solar power. Located in an evergreen forest near a snowcat barn and snowmobile road, which was active in the winter. |
| Snodgrass | 38.9271°N, 106.9905°W | 3333 m | Oct. 2021-July 2023 | Instrumentation was mounted on scaffolding and ran on solar power. Remote, off-trail location on the side of Snodgrass Mountain, but directly north of Crested Butte town. |
| CBTop | 38.8888°N, 106.9450°W | 3482 m | June 2022-July2023 | Instrumentation was mounted on shared tower and ran on solar power. Located near the top of Crested Butte Ski Resort. |

**Table 1.** Basic description and details for each of the six sites in SAIL-Net.




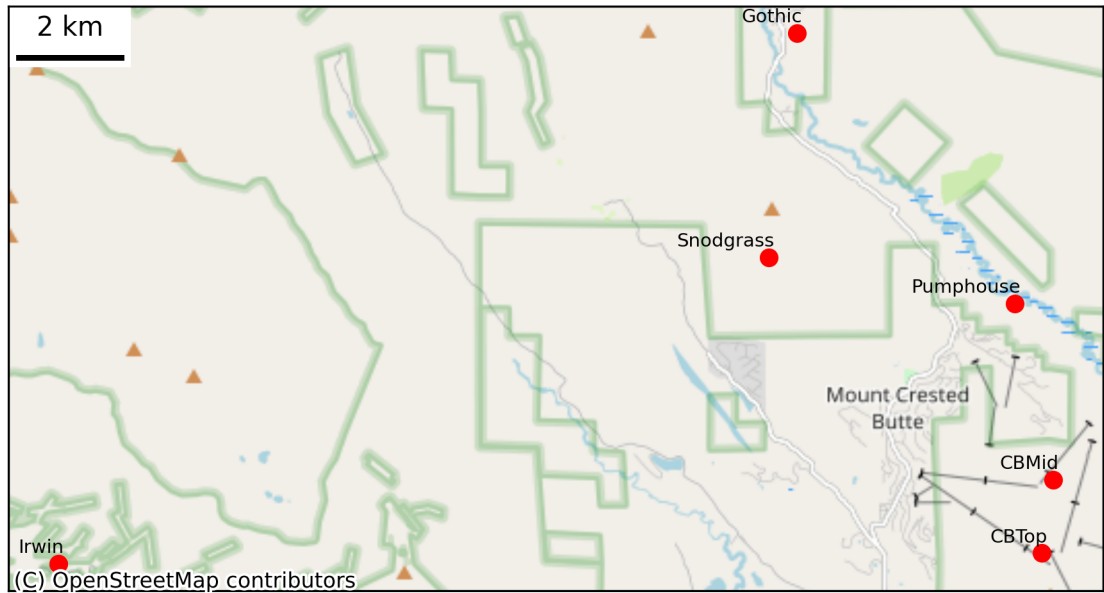

**Figure 1.** The six sites in SAIL-Net are all marked with a red dot. The network spanned approximately 8 km vertically (North-South) 14 km horizontally (East-West), and covered approximately 750 m of elevation difference. ©OpenStreetMap contributors 2024. Distributed under the Open Data Commons Open Database License (ODbL) v1.0.

until drift was observed by the PSL check. We also assumed that the PSL checks were representative of an entire month. Lastly, if a month missed a PSL check, we assumed the drift was linear to allow interpolation between missing PSL checks.

The post-correction process involved shifting the boundaries of bins to size 500 nm PSL in the correct bin at the completion of the data correction process. A POPS experiences drift for two primary reasons: either the laser diode loses intensity over time or the mirror that reflects light becomes dirty. In either case, the lower intensity of light causes particles to be sized smaller than their true size. The digitizer in the POPS reads this raw signal of the light intensity. The sizing range of the POPS is determined by taking the base 10 logarithm of the range of the digitizer. In logarithmic space, the range is 1.75 to 4.806. The bins of the

POPS are then determined by dividing this range into $n$ bins of equal width $w$, where

$$w = (4.806 - 1.75)/n. \tag{1}$$

These log values are converted to diameter space using Mie theory. In diameter space, the bins are no longer equal in width.

The intuition for the post-correction comes from considering the raw signals that the digitizer would receive and scaling the signal to properly bin it. The following explanation shows that this is equivalent to simply shifting the current bins of the POPS.




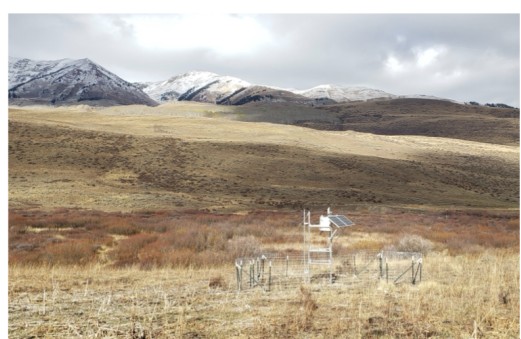
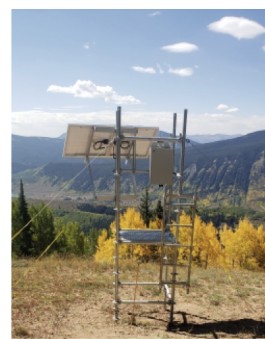
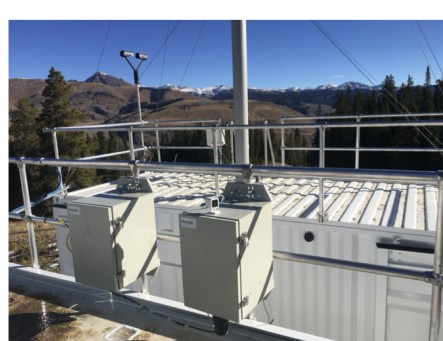

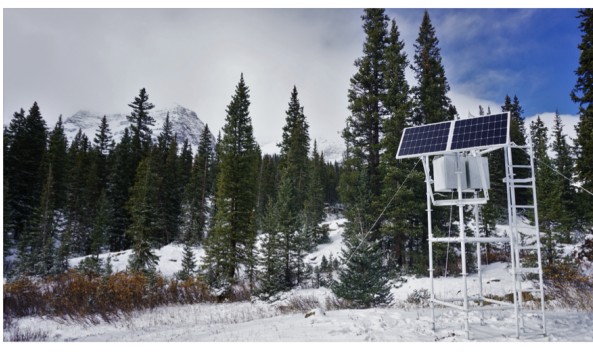
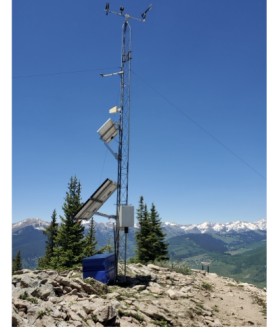
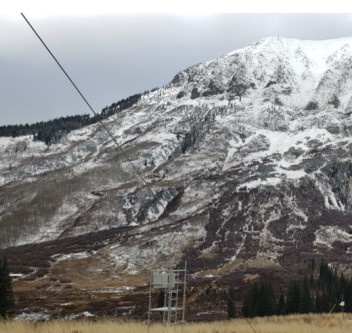

**Figure 2.** Photos of the six sites in SAIL-Net. From left to right, top to bottom, the sites are Pumphouse, Snodgrass CBMid, Irwin, CBTop, and Gothic.

When the POPS sizes particles accurately, 500 nm PSL should be placed into the bin containing 500 nm sized particles. Let the midpoint of this bin in logarithmic space be called $x$. Suppose 500 nm PSL is instead sized into a different bin with midpoint $y$ in logarithmic space. Thus, the digitizer saw a raw signal of $10^y$ instead of $10^x$. To correct for this error, we would need to scale all raw signals by $10^x/10^y$. Since this is a post-correction and all raw signals have already been received, we instead scale all digitizer bin boundaries by $(10^x/10^y)$ so that the drifted signals would be binned properly. The bin boundaries, $b_i$ are defined

in logarithmic space using the range of the digitizer and Eq. 1, but can be converted to raw signal using $10^{b_i}$. Thus, to account for the drift, we apply a shift to all bin boundaries: $10^{b_i}(10^x/10^y)$. To then convert the raw signal back into logarithmic space, which is necessary for converting back to diameter space, we take the base 10 logarithm of the raw boundaries:

$$\log_{10}(10^{b_i}(10^x/10^y)) = b_i + (x - y). \tag{2}$$

Since $x$ and $y$ are the midpoints of *equal* sized bins in logspace, $(x-y)$ is equivalent to the width of a bin, $w$, times the number

of bins apart they are, $m$. Thus the post-correction ends up being as simple as shifting all bins up by $m$ spaces, keeping bin boundaries the same, until 500 nm PSL, and all particles, are sized correctly. Because of this upward shifting, the lowest size that the POPS measured increased throughout the deployment. For the majority of the following analysis, the minimum particle size used will be 170 nm instead of the 140 nm that is standard with the POPS to account for this shift.





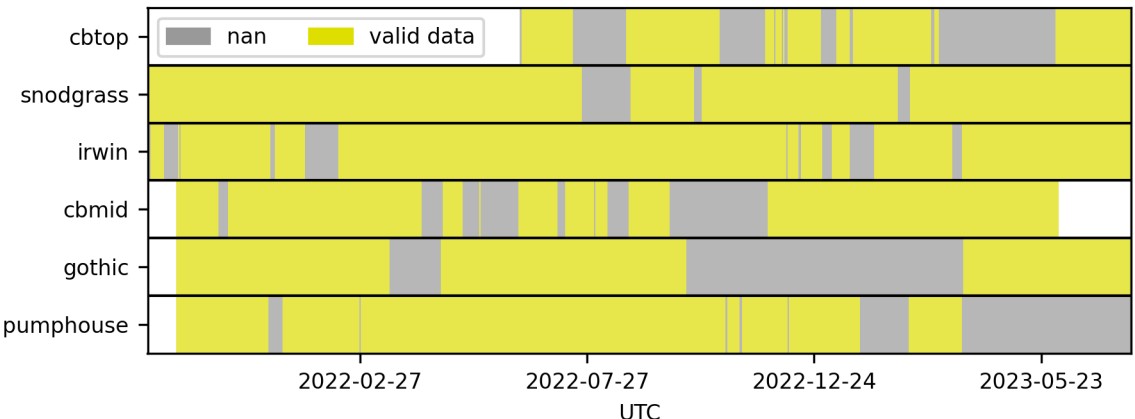

**Figure 3.** Plots of the completeness of 170 nm-3.4 $\mu$m POPS data. The gray squares indicate times that the site was in place but no data was recorded, or the data did not meet quality assurance standards. The yellow squares indicate days that the site has valid data. The percent of valid data for each site is cbtop, 60%; snodgrass, 92%; irwin; 88%; cbmid, 76%; gothic, 66%; and pumphouse, 75%.

Once data were rebinned, additional smoothing was performed by computing one minute rolling averages of the data to
remove excessive noise. These post-corrected and cleaned data were used for all analysis described in the following section.
Figure 3 displays the completeness of 170 nm-3.4 $\mu$m size particle data for each site in the network. We assume that the
post-correction process has removed any instrument caused variation between different POPS, and therefore, the remaining
variability observed in the data is due to environmental conditions. These cleaned POPS data, along with raw POPS data and
ClouckPuck data, are also publicly available on the ARM Data Discovery (Gibson and Levin, 2023). INP data will become
available once the filters have been analyzed by Perkins et al. (2023).

## 3   Results and discussion

This section uses data from the six POPS to address the science questions proposed in the Introduction. The POPS produced the
longest and highest temporal resolution dataset, which allows the study of spatiotemporal variability in aerosol concentrations
and distributions. Figure 4 displays the complete time series of 170 nm to 3.4 $\mu$m sized aerosol concentration data from the
POPS at the six sites. The data are averaged daily by UTC.

The daily averaged data indicate that all the sites exhibited similar daily behavior and seasonal trends. The sites experienced
higher total aerosol concentrations in the summer and lower in the winter, which was consistent with the seasonal trends of
other mountainous regions (Gallagher et al., 2011). Concentrations peaked in the later summer and reached a minimum in
January. The maximum recorded concentration occurred on June 13, 2022, at Gothic, with an average daily concentration
of 672 cm$^{-3}$ due to smoke from the Flagstaff wildfires burning in Arizona. Concentrations were again abnormally high in





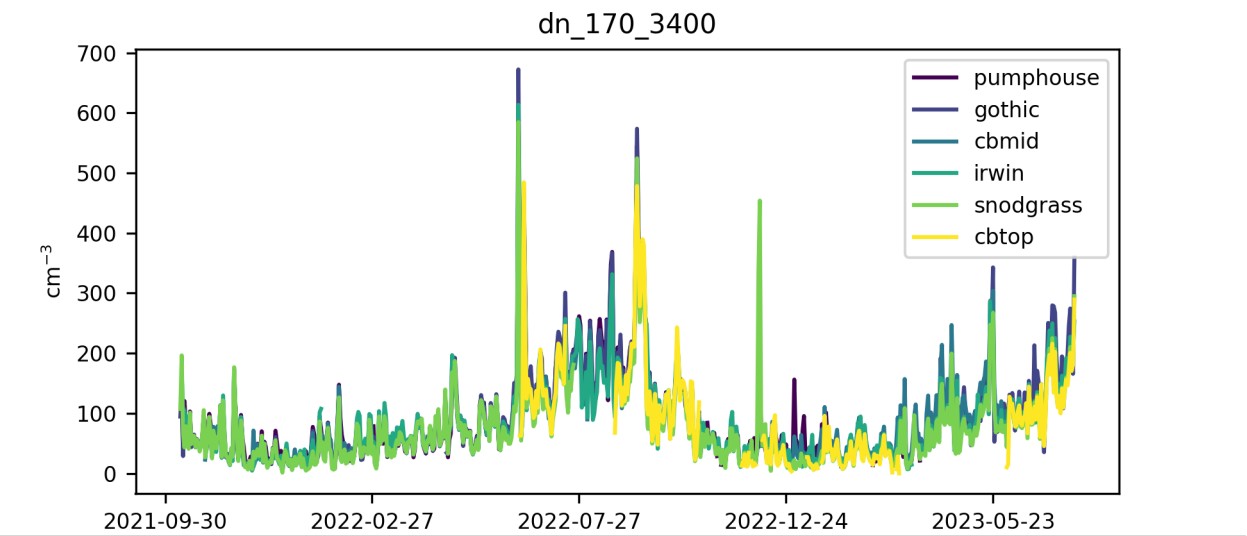

**Figure 4.** The time series of daily averaged concentration of 170 nm to 3.4 $\mu$m sized aerosol for the six sites in SAIL-Net.

September 2022 due to biomass burning as well. The unique differences and trends in the data are discussed below and broken into three sections based on the science questions posed in the Introduction.

### 3.1 Seasonality and diurnal patterns

The POPS data experienced both seasonal and diurnal cycles. In this section, we use the time series of the network mean of
the data to study the temporal variability of aerosol. The network mean at time $t$, $N_t$, is the average of $m$ sites' values at time
$t$. Thus given a time series at each site; $\{x_{i,t=1}, x_{i,t=2}, \ldots, x_{i,t=n}\}$ where $i$ is the site number, the network mean timeseries of
$m$ sites is

$$\{N_1, N_2, \ldots, N_n\} = \left\{ \frac{\sum\limits_{i=1}^{m} x_{i,1}}{m}, \frac{\sum\limits_{i=1}^{m} x_{i,2}}{m}, \ldots, \frac{\sum\limits_{i=1}^{m} x_{i,n}}{m} \right\}. \tag{3}$$

Since the network mean takes an average of spatially dispersed sites, it removes much of the noise and variability caused by
local sources or instrument drift and can be used as a proxy for a model grid cell in the region.

Most sites had gaps in data at some point, so when one or multiple sites were missing data, the network mean was computed from the sites with data. This choice was made to preserve as much temporal coverage as possible and attain a clear picture of seasonal trends. For further discussion of the network mean and its ability to represent the East River Watershed, see Sect. 3.3. In the following analysis, the sum of aerosol concentrations between 170 nm and 3.4 $\mu$m are used, unless otherwise specified.
SAIL-Net collected data during two very different winters. The 2022 snowpack in the Gunnison Basin, which the ERW is a part of, ended up being close to the median for the region. However, if it were not for a large snow in late December 2021,





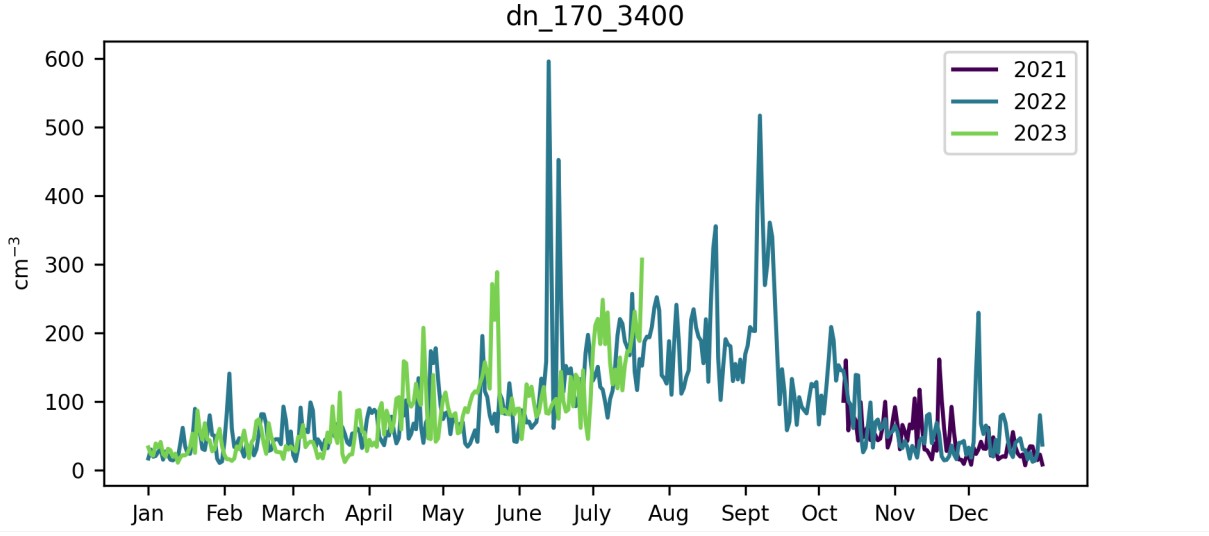

**Figure 5.** The network means of daily 170 nm to 3.4$\mu$m sized aerosol concentrations by day of the year.

the snowpack would have been well below normal. In contrast, the 2023 winter saw higher than normal snowfall, with snow water equivalent peaking in the 90th percentile of the 30-year median (NRCS, 2023). Despite the very different winters, the daily average aerosol concentrations for 170 nm to 3.4 $\mu$m sized particles of the network mean had similarities over the years.

Figure 5 displays the network means overlaid by days of the year. The main difference between the two years occurred on 13 June 2022, when the spikes in concentration were due to smoke from the Flagstaff wildfires in Arizona. The maximum recorded concentration occurred during this time. The minimum recorded concentration of the network mean occurred on 24 December 2021, with a concentration of 7 cm$^{-3}$. This minimum was likely caused by scavenging from heavy snow that fell on the same date. Below, we further analyze the temporal trends in aerosol data.

All SAIL-Net sites experienced diurnal cycles in aerosol concentration, but these cycles changed throughout the year. Figure 6 plots the average diurnal cycle of the network mean for each month of the SAIL-Net collection period. For this analysis, we removed data from mid-June 2022 so that the abnormally high concentrations caused by wildfire smoke would not affect the trend. The times in this plot have been converted to local time to allow for easy viewing of the day's effect on aerosol concentrations. When SAIL-Net ran for the same month of different years, both years are displayed on the same plot. All seasons

but winter have clear diurnal cycles, where aerosol concentrations decrease during the day and increase overnight. This pattern was more distinct in the warmer months, where total concentrations rapidly decreased at sunrise and began increasing close to sunset. The less obvious diurnal cycles in the winter months could partially be attributed to less vertical mixing of the boundary layer throughout the day (Gallagher et al., 2011).

These results were partially consistent with the diurnal analysis from Gallagher et al. (2011) at Whistler Mountain, which

studied the seasonal and diurnal patterns of CCN. They found that diurnal cycles were more distinct in warmer months and



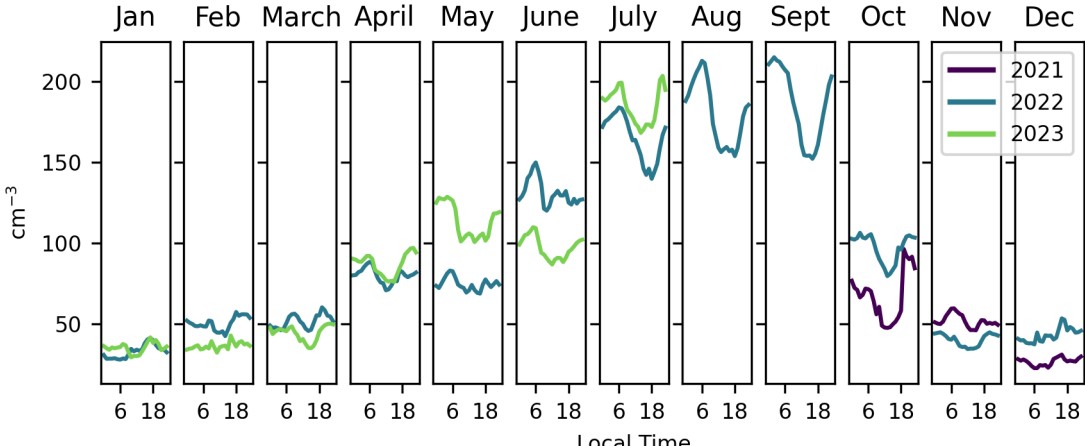

**Figure 6.** The daily diurnal cycle of the network mean averaged monthly and overlaid by year for each month of SAIL-Net. Times have been converted to local time for ease of interpretations.

less so in the winter. They also observed a small dip in CCN concentrations around sunrise but detected increasing CCN concentrations from 08:00 until approximately 16:00 LST as a result of new particle formation. This daytime increase was not observed in the SAIL-Net data. We hypothesize this was because the POPS cannot detect small enough particles to observe new particle formation. The small increase seen around 18:00 LST most months in Fig. 6 may be a signal from particles that have grown large enough to be detected by the POPS. Observations from Hallar et al. (2011) at Storm Peak Laboratory in northwestern Colorado saw new particle formation begin around noon local time in the winter months, so it is likely this occurred at a similar time in the ERW.

The diurnal cycles of the months containing two years of data had remarkably similar shapes for the most part, highlighting the consistent impact that the seasons and daytime have on aerosol concentrations. In most months, the diurnal cycles mainly differ by a scaling factor, indicating that the main difference between the two years was the concentration of aerosol. The distribution of particle sizes changed monthly and also differed between the two years. Figure 7 displays the average monthly particle size distributions overlaid by month. In general, supermicron concentrations peaked in April and were higher in March through June, primarily due to aeolian dust transported from the desert southwest (Skiles et al., 2015). However, the two spring dust seasons were noticeably different. According to the POPS data, supermicron concentrations increased in March and lasted through June 2022, whereas in 2023, April saw the highest supermicron concentrations. Submicron concentrations peaked in the summer and quickly dropped off in the fall. This behavior was also apparent in Fig. 6 since submicron particles dominate the bulk of the number concentration.

Figure 8 plots the time series of daily averaged particle size distributions for the entire measurement period. Here, we see the seasonality in different particle sizes. Particles between 140 and 300 nm increased in the spring and early summer and peaked in late summer. There was a period in both winters around late December and early January when the air was extremely clean,



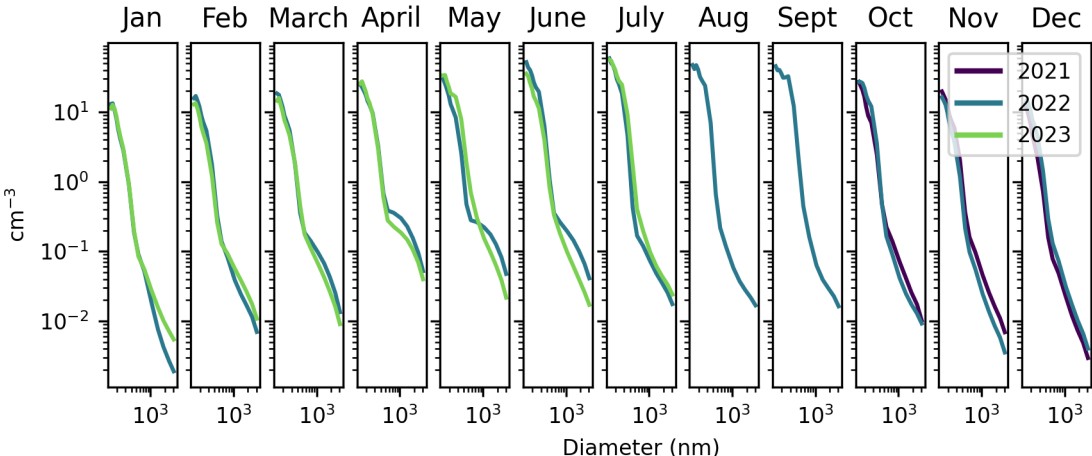

**Figure 7.** The particle size distribution of the network mean averaged monthly for each month of SAIL-Net. These plots used the full 140 nm to 3.4 $\mu$m size range of the POPS.

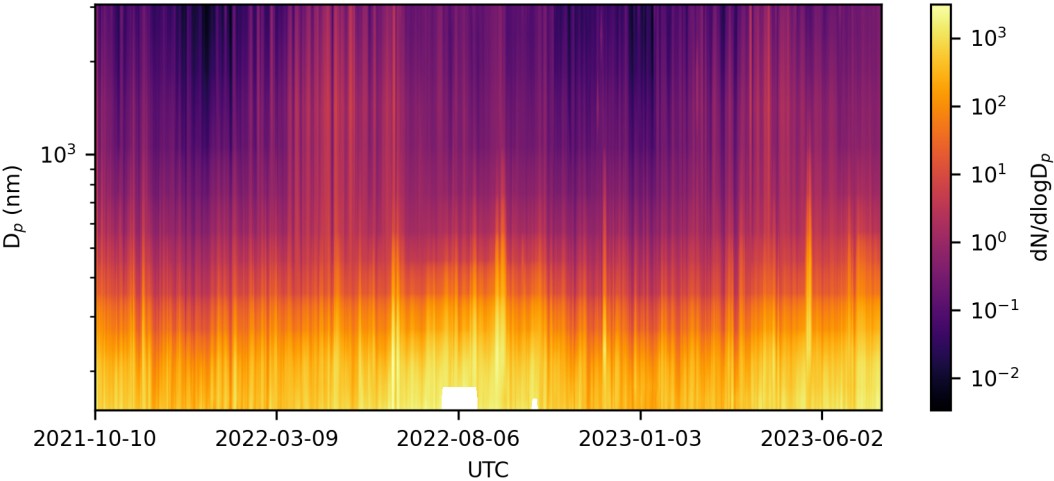

**Figure 8.** The time series of the measured particle size distributions of the network mean. Data were averaged daily. This plot uses the full 140 nm to 3.4 $\mu$m size range of the POPS.

and there were very few particles larger than approximately 300 nm. This figure also provides another look into the spring dust events, which were characterized by higher than normal concentrations of supermicron-sized particles.





## 3.2 Spatial variability

Networks of sensors are useful in cities and more polluted areas because aerosol concentrations vary dramatically over small

spatial scales (Popoola et al., 2018; Caubel et al., 2019). In less populated areas such as the ERW, there are not as many local

sources of emissions. However, aerosol properties can vary with elevation changes (Zieger et al., 2012). This section explores

the spatial variability of the region and its relationship with elevation.

Figure 4 showed that all sites were reasonably similar on a daily timescale. However, there was still variability within the

data, especially on a smaller timescale. Sub-daily variability was primarily due to local emissions and distances between sites.

We were able to identify the sources of some of this variability, and a few examples are described below.

CBMid and Irwin experienced spikes in 155 nm-300 nm sized particles from late November to early April, which we at-

tributed to nearby snowcat and snowmobile activity. The top plot in Fig. 9 demonstrates this for a few days of Winter 2022. The

spikes at CBMid occurred during the night local time, corresponding with Crested Butte Ski Resort's nightly grooming of their

runs. The spikes at Irwin occurred roughly between 9 am and 3 pm local time, corresponding with the times that snowmobiling

and other winter activities would take place. Concentrations at Gothic were influenced by increased anthropogenic activity in

the summer. The middle plot of Fig. 9 displays these effects compared to Pumphouse, which was also in the East River Valley.

The road to Gothic opened at the end of May 2022, aligning with the start of noisy spikes occurring at Gothic. It is unclear if

these spikes were due to road traffic or other activities near the town of Gothic, such as campfires. Variability between sites

was also due to their dispersed locations. The bottom plot of Fig. 9 displays this behavior on 13 June 2022 when smoke from

the Flagstaff Wildfires blew into the region. In this case, the sites report similar concentrations at a lag of one another, leading

to increased variability as the plume moved into the area.

Beyond variability caused by local sources, we found that the variability between sites was partially influenced by their dif-

ferences in elevation, supporting the findings from Zieger et al. (2012). Figure 10 plots the average pairwise percent difference

in aerosol concentrations between two sites as a function of the elevation difference between the sites. The percent difference

was calculated daily as the absolute difference between the two sites divided by their average. These daily errors were then

averaged over the total SAIL-Net deployment period to attain the plots in Fig 10. This was done for three groupings of particle

sizes: 170 nm-300 nm, 300 nm-870 nm, and 870 nm-3.4 $\mu$m. These groupings were chosen based on the size ranges of particles

that consistently had more similar concentrations. The r-value of 0.48 for 170 nm-300 nm sized particles indicates there is a

positive linear correlation between these two variables. For this size range, the most similar sites were Pumphouse and Gothic,

with an average difference of 13.4%. Gothic and Pumphouse were both located in the East River Valley and were the two

lowest elevation sites. The most different sites were the two geographically closest sites, CBTop and CBMid, with an average

difference of 35%. There was no such correlation for the other groupings.

The average pairwise percent difference in aerosol concentration as a function of geographic distance between the two

sites all yielded a negative correlation on average. This result indicates that the common assumption that spatially close data

are more similar does not apply here. These findings suggest that the variability between sites was partially due to their

elevation differences for 170 nm to 300 nm sized particles. Section 3.3 discusses the relationship between site concentrations





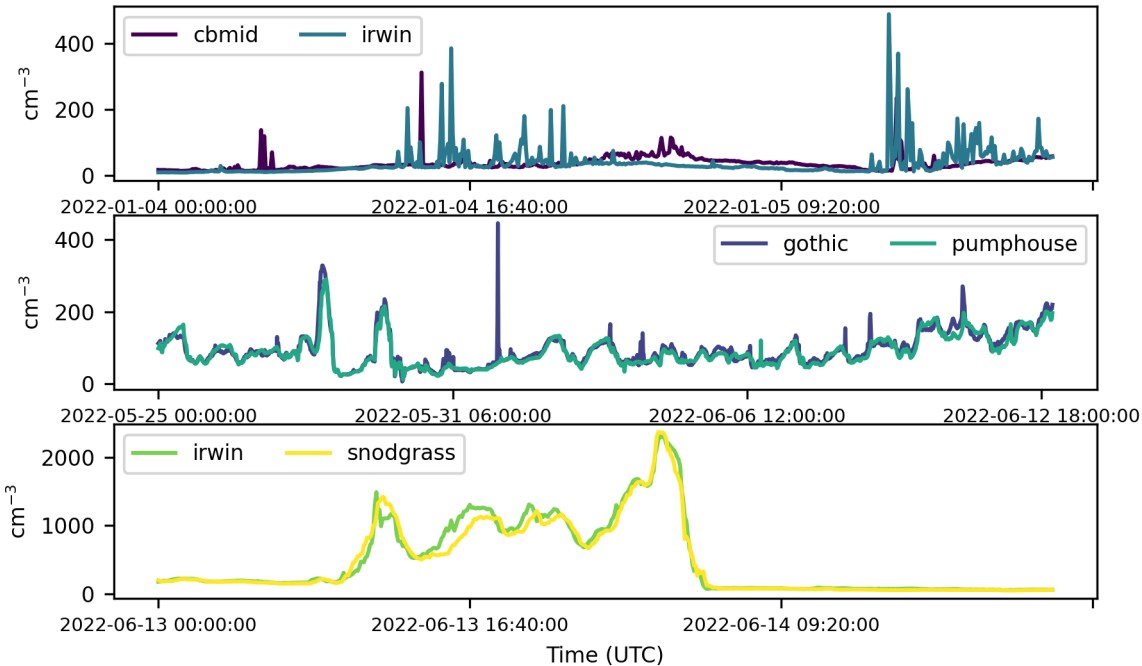

**Figure 9.** Examples of sub-daily variability among SAIL-Net. The top figure displays spikes in 155 nm-300 nm sized aerosol for CBMid and Irwin, which were both affected by winter snowsport activities. The middle plot displays noisy spikes at Gothic which began after Gothic road opened for the season. The bottom figure displays a lag in total aerosol concentration when a smoke plume moved into the region on 13 June 2022.

and elevations further. The remaining variability in the data was likely due to the unique ecological and local differences at each site.

The variability across the sites also changed seasonally. Figure 11 plots the coefficient of variation (CV) of the sites over
time. The CV represents the dispersion within a set of data. Here, the data were grouped by time, so that each time step provided a set of data for which to compute the CV. Each set was normalized using min-max scaling before computing the CV. This choice was made to account for the seasonality of the data while maintaining the relative distance between values.

Based on our results, there was less variability among the sites during the summer of 2022 than in other seasons. The variability also began trending downward as the weather warmed in 2023 but then increased in the last few weeks of deployment.
We hypothesize that the increased variability in the cooler seasons could be partially due to the impact of snow-covered ground on the daytime convective boundary layer. Adler et al. (2023) saw a low convective boundary layer over snow-covered terrain in the East River Watershed and observed inversions at night. In some observations, the boundary layer was low enough that some high-elevation sites in SAIL-Net would be above the boundary layer, and thus measure different aerosol concentrations



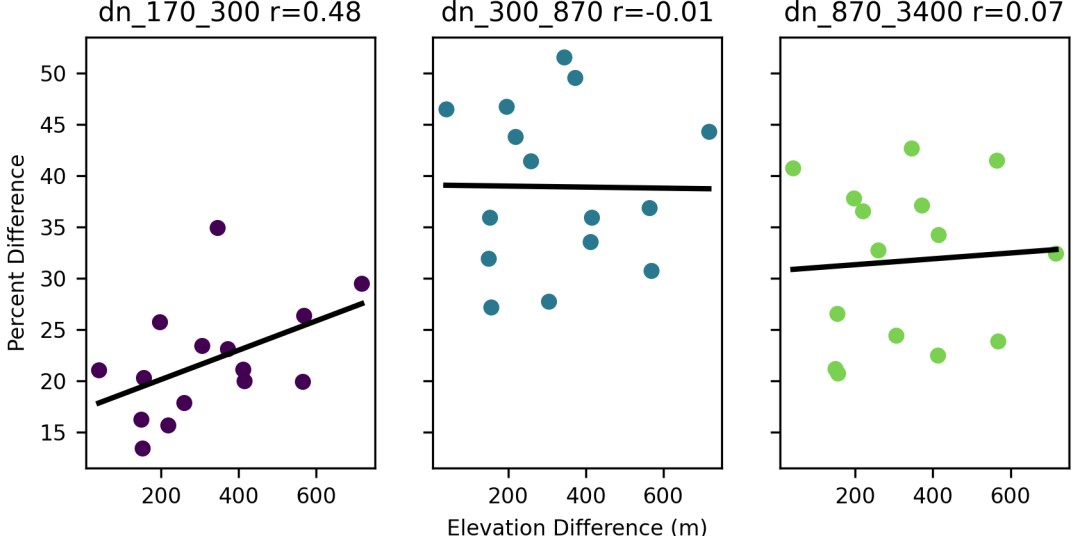

**Figure 10.** The average percent difference between pairs of sites as a function of the elevation between them. The average percent difference was computed from daily averages, and then averaged over the entire SAIL-Net deployment period to attain these values.

than below the boundary layer. However, another factor that likely affected the higher variability in the winter months was the

low aerosol concentrations across the sites. The depths of winter experienced concentrations of less than $100\,\mathrm{cm^{-3}}$ on average. In these clean conditions, any local variability would amplify the differences between sites.

### 3.3 Network representation

The previous subsections highlighted the temporal and spatial variability in the ERW. We now use these data to investigate the optimal network design in the region and determine if a single site can accurately represent the aerosol properties of the region.

As defined and studied by Schutgens et al. (2017), the representation error is the ability of a measurement to represent a larger area. There is often a significant difference between model estimates for a region and observed point measurements, leading to inaccuracy (Schutgens et al., 2016). The representation error quantifies how similar each site is to the network mean (Eq. 3 in Sect. 3.1). Local sources affect measurements at a single site, so it can be advantageous to average over multiple sites to gain a proxy for the region. However, as explored in Sect. 3.2, there was an underlying structure to some variability. The

representation error quantifies how different a single site is from the network mean and provides meaningful insight into the usefulness of a network of sites in complex terrain.

Using the equation from Asher et al. (2022), the representation error, $e_t$, is the normalized difference between a site observation and the network mean for an averaging period $t$

$$e_t = \frac{O_t - N_t}{N_t}. \tag{4}$$





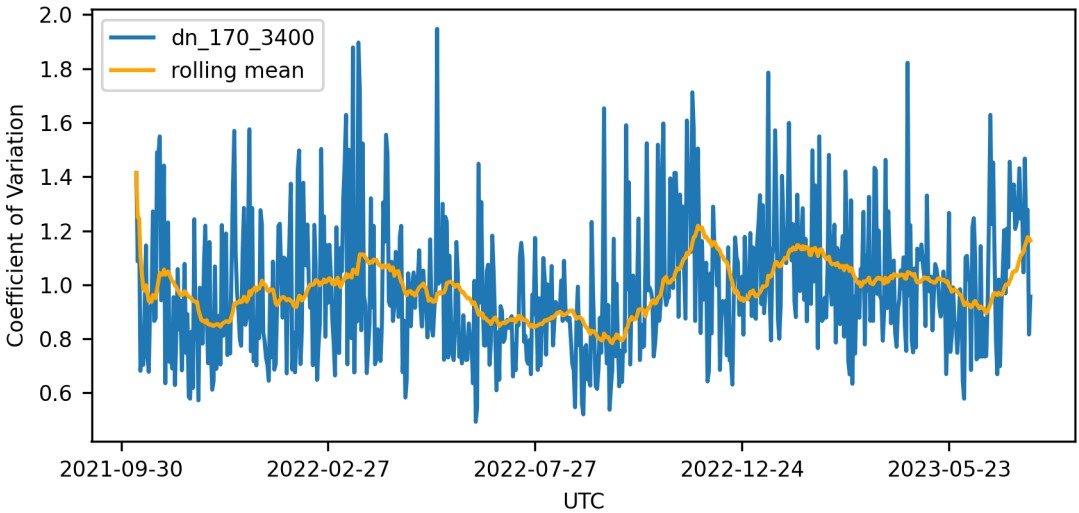

**Figure 11.** The time series of the coefficient of variation of daily averaged 170 nm to 3.4 $\mu$m aerosol concentration for the SAIL-Net sites. The 30 day rolling average removes much of the noise from the data. There was less variability amongst the sites in summer 2022.

During POPS-Net in the Southern Great Plains, Asher et al. (2022) found the representation error decreased when data were averaged over longer periods. This was true for SAIL-Net as well. We will use daily averaged data for the following analysis.

   Figure 12 plots the daily averaged representation error for the six sites using three groupings of aerosol size ranges. A value closer to zero indicates that site observation represents the region well. The representativeness of the sites was typically worse in the winter than in the summer, as indicated by representation errors farther from zero. This could be attributed to the

increased variability across the sites in the winter, which we saw with the coefficient of variation in the previous subsection. This seasonality in the representation error indicates that a single site alone would not consistently represent the region with the same accuracy throughout the year.

   Figure 13 is the result of averaging over all days of data in Fig. 12 and plotting the average and the range for each site and size range. The most representative site should have a mean close to zero and a small range. However, for these data, no single

site had all three size ranges with an average closest to zero and smallest range. Gothic was the most representative site for particles in the size range of 170 nm to 300 nm, CBTop was the most representative for particles from 300 nm to 870 nm, and Pumphouse was the most representative site for particles in the size range of 870nm to 3400 nm.

   To determine the most representative site for all size ranges, we assigned a score to each size range by summing the range and absolute value of the average. For each site, we summed the scores over the three size ranges. The site with the lowest

score was deemed the most representative overall. We computed the representation errors for each season and applied this scoring approach to study how representation changes. Table 2 displays the ordering of most representative sites for each season that SAIL-Net recorded. The most representative site was inconsistent over time, suggesting there was not a single most





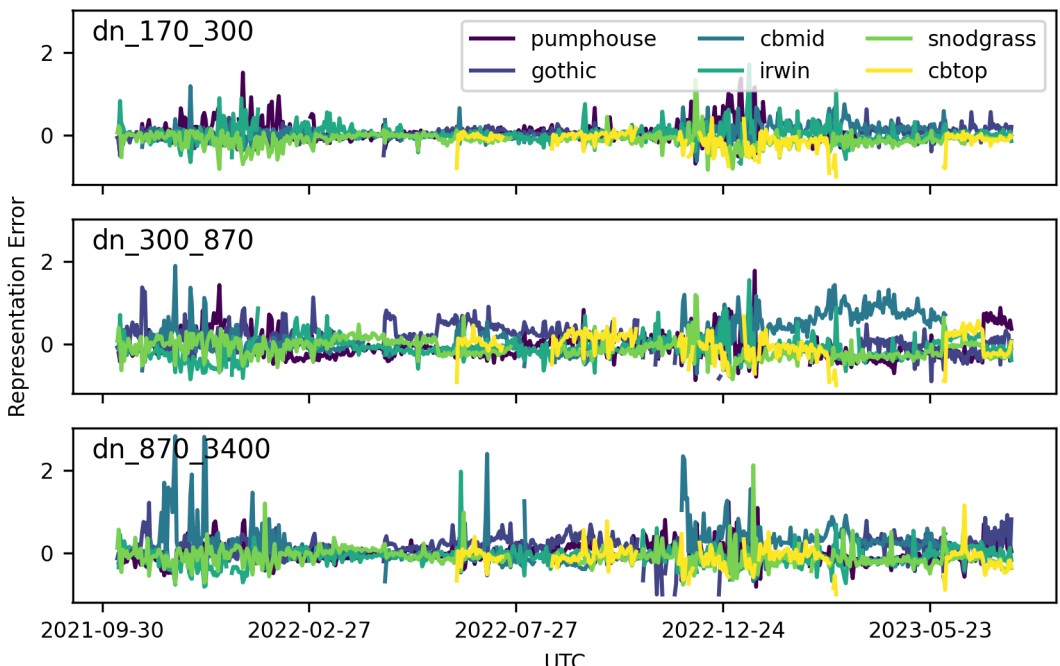

**Figure 12.** The time series of the daily representation error for the six SAIL-Net sites, broken down into three size ranges: 170 nm-300 nm, 300 nm-870 nm, and 870 nm-3.4 $\mu$m. The representation was worse in the winter.

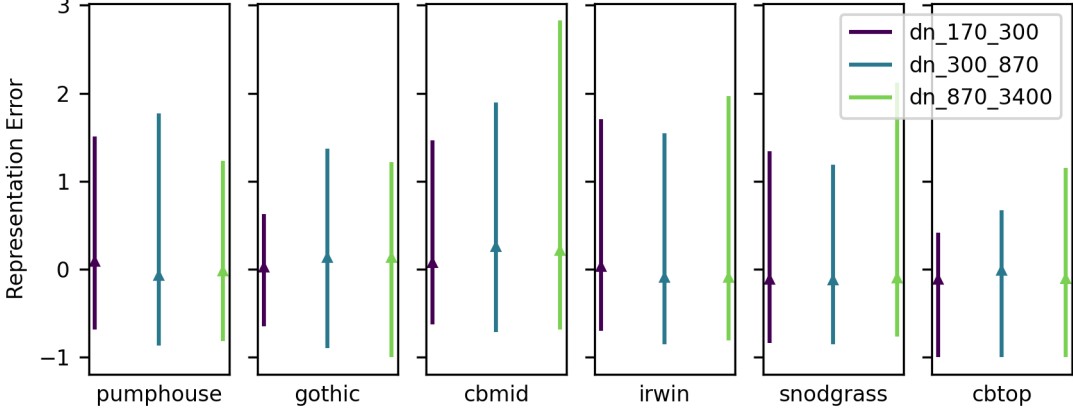

**Figure 13.** The mean and range of the representation error computed by averaging over the daily representation errors. The data are broken down into the three size ranges: 170 nm-300 nm, 300 nm-870 nm, and 870 nm-3.4 $\mu$m.





representative site throughout the seasons. Furthermore, there does not appear to be any pattern in the elevations or locations of sites that are the most representative. In Winter 2022, the most representative site was Gothic, located in the East River Valley, while the following winter, the most representative site was CBTop, located near the top of Crested Butte Mountain. Irwin and CBMid were also never marked as the most representative sites. Given that Irwin was the most isolated site, located farther west than the rest, this was not too surprising. However, we do not yet have an explanation for this behavior at CBMid.

| | **F21** | **W22** | **SP22** | **SU22** | **F22** | **W23** | **SP23** |
|---|---|---|---|---|---|---|---|
| **First** | Snodgrass | Gothic | Pumphouse | Snodgrass | Gothic | CBTop | Snodgrass |
| **Second** | Pumphouse | CBMid | Snodgrass | CBTop | Irwin | Snodgrass | Irwin |
| **Third** | Irwin | Snodgrass | Gothic | Pumphouse | CBTop | Irwin | Pumphouse |
| **Fourth** | Gothic | Irwin | Irwin | Gothic | Pumphouse | CBMid | CBMid |
| **Fifth** | CBMid | Pumphouse | CBMid | Irwin | Snodgrass | Pumphouse | Gothic |
| **Sixth** | - | - | - | CBMid | CBMid | - | CBTop |

**Table 2.** The ordering of the most representative sites for each season. CBTop was installed on 14 June 2022, explaining the lack of a sixth site for the first three seasons. The POPS at Gothic was broken during the winter of 2023. Spring 2023 also includes the last month of data from 20 June - 22 July 2023.

This representation analysis quantified the ability of a single site to represent the larger area, but given the varying elevations of the sites, we also explored how representative the sites were of the vertical profile of air in the region. The six SAIL-Net sites were intentionally placed at various elevations to span a portion of the vertical profile of altitudes in the ERW. To quantify how representative these sites were of the vertical profile in the region, we compared our data to the data collected during tethered balloon system (TBS) flights that took place in the region during the SAIL campaign (Mei et al., 2023). The TBS flights occurred at Gothic in 2022 and at Pumphouse in 2023. Each balloon was equipped with a POPS from Handix Scientific, which allowed for easy comparison with the POPS at the SAIl-Net sites.

This representation analysis quantified the ability of a single site to represent the larger area, but given the varying elevations of the sites, we also explored how representative the sites are of the vertical profile of air in the region. The six SAIL-Net sites were intentionally placed at various elevations to span a portion of the vertical profile of altitudes in the ERW. To quantify how representative these sites were of the vertical profile in the region, we compared our data to the data collected during tethered balloon system (TBS) flights that took place in the region during the SAIL campaign (Mei et al., 2023). The TBS flights occurred at Gothic in 2022 and at Pumphouse in 2023. Each balloon was equipped with a POPS from Handix Scientific, which allowed for easy comparison with the POPS at the SAIl-Net sites.

During every TBS flight, the balloon was sent up vertically through the atmosphere. The balloon remained approximately in the same geographic location so that each flight generated a profile of the vertical air column in the region. To compare the data from the TBS flight with SAIL-Net, we built a "vertical column" from the SAIL-Net sites. The "vertical column" was built by associating the site's elevation above sea level with its measured total aerosol concentration, ignoring the geographic location of the site. We then computed the error between the concentrations reported by the POPS on the TBS flight and each site by





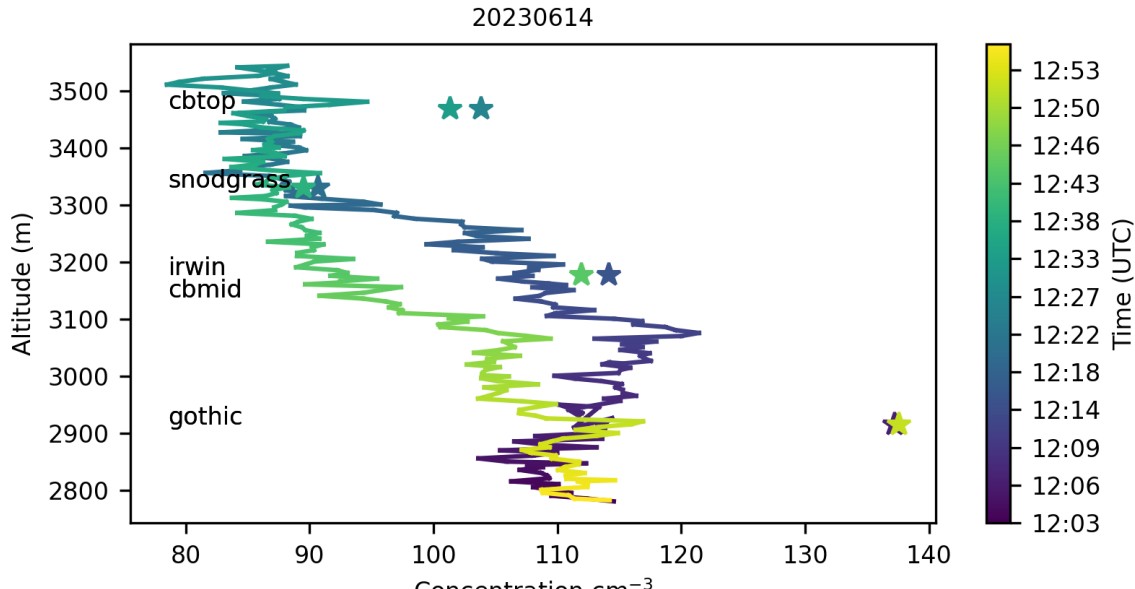

**Figure 14.** The POPS data from the TBS flight on 14 June 2023, plotted with the measurements at each SAIL-Net site. When the elevation above sea level of the balloon passed within 2.5 m of the elevation of the SAIL-Net site, the mean of the concentration at the site was computed, and this is the value plotted by the stars.

comparing the concentrations at times $T$, where $T$ is a set of times determined by when the balloon passed within 2.5 m of the altitude of the given site. Figure 14 shows an example of TBS flight data plotted with the SAIL-Net site data.

For each flight, we computed both the absolute value of the percent difference between the vertical column and the site and the absolute difference between the two. This choice was made because the total aerosol concentrations were so low in the winter that the percent error could become a less useful metric for understanding the differences in the region over time. Once the percent error and absolute difference were computed for each site during the flight, we computed the median of these differences for all the flights deployed on the same day. We chose the median because we wanted to obtain a metric that represented the typical difference between the site and vertical column and did not want to be influenced by outliers which were sometimes present. To then obtain a value that represented the difference between the complete vertical column from the flight and the "vertical column" from the sites, we computed the mean and median from the errors of the sites for each day.

Figure 15 plots these statistics for each day that flights occurred. The means of both plots were skewed by outliers, so we believe that given our small sample size of at most six sites, the median is a better measurement of the error between the TBS vertical column and SAIL-Net's "vertical column". Like the results of the previous subsection and the representation error analysis, the percent error and absolute error were overall larger and more dispersed in the cooler months of January and April 2023, indicating more variability in the region. Differences between measurements, potentially caused by local sources and/or a low convective boundary layer, were further amplified due to the low total concentrations recorded in the winter.



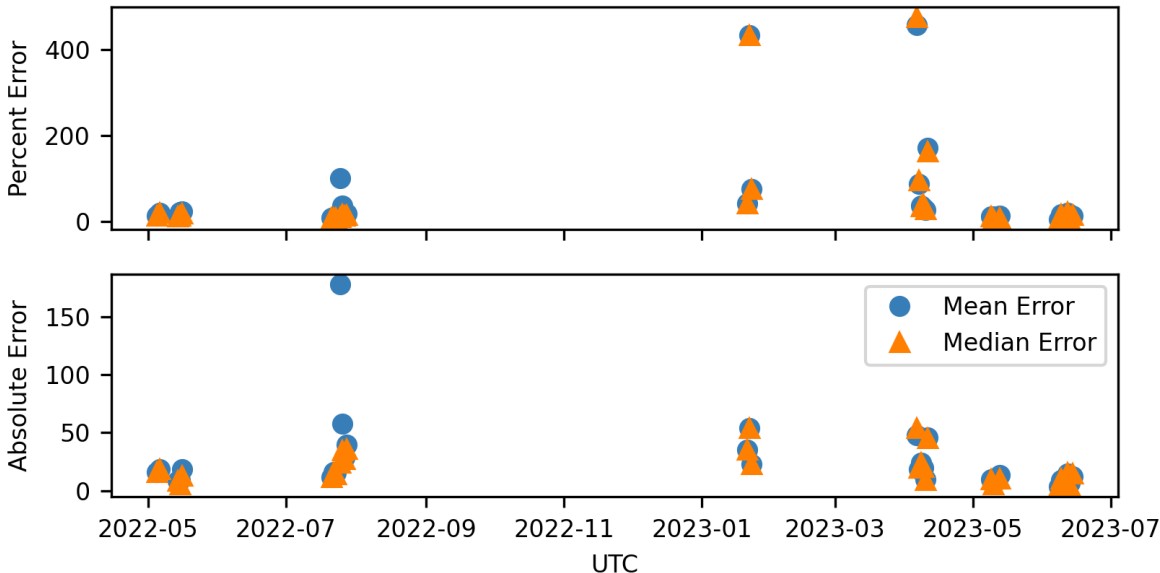

**Figure 15.** The mean and median of the absolute percent errors and absolute errors computed daily from the TBS flights.

There was a significant decrease in the errors between April and May 2023. This could be due to the spring awakening and increase in human activity raising total aerosol concentration in the region. May 2022 and June 2023 also had low errors. July 2022 was the observed exception to this trend of warmer months having lower errors. Numerous wildfires were burning in the Southwestern US during this time, so the increased errors could potentially be due to the variability of these plumes across the region. Overall, the sites better represented the vertical profile in the ERW in warmer months, with the lowest median percent error of 5.4% occurring on 16 June 2023.

Based on this analysis, there was no single site that best represented aerosol concentrations in the ERW. However, the errors tended to be low enough in the summers that any single site could be a sufficient approximation of the region, depending on a user's desired error tolerance. The winter months posed more of an issue since any localized sources or changes in the daytime convective boundary layer could drastically decrease the ability of any site to represent the region.

## 4 Conclusions

SAIL-Net was the first of its kind in mountainous terrain and now presents a complete dataset highlighting the spatiotemporal variability of PM2.5 in complex terrain. SAIL-Net observed seasonal and diurnal cycles in aerosol concentrations. The highest concentrations occurred in late summer, but supermicron concentrations peaked in the spring, likely due to aeolian dust. Diurnal cycles were more pronounced in warmer months, agreeing with the findings of Gallagher et al. (2011). There was more variability between the sites in the winters than in the summers, possibly because the lower concentrations in the winters





caused sites to be more sensitive to local sources. There is also a possibility that the winter time convective boundary layer
was low enough that some higher elevation SAIL-Net sites were above it, also leading to increased variability, but more work
should be done here to determine if this is true.

The differences in concentration between the sites were partially related to their elevations, with an R-value of 0.48 relating
elevation proximity to measurement similarity for 170 nm to 300 nm sized particles. This relationship between concentration
and elevation was further supported by the ability of the sites to represent the vertical profile of air in the region. From the
comparisons between site data and TBS flights, the error in the sites representing the vertical profile of air in the region was
as low as 5.6% in June 2023. The variability between sites was inconsistent over different seasons, underscoring the potential
inadequacy of a single site to consistently represent the complex terrain in the ERW. However, the similar daily trends across
the sites indicate that on a daily timescale, there is minimal variability in the region. Compared to the range of representation
errors seen by Asher et al. (2022), SAIL-Net sites did experience larger representation errors over a smaller spatial region. This
result emphasizes the increased variability in complex terrain and also supports the findings from Zieger et al. (2012) in the
Swiss Alps.

There is future work with these data that could be done. While this manuscript focused on the analysis of the data, there
are opportunities for modeling and further analysis of the data. One such direction would be combining these data with other
observations to begin to explain the behaviors observed here. For example, one could explore the possible causes of increased
variability in the winter. Another direction would be exploring the diurnal cycles in aerosol concentrations to understand why
concentrations decrease during the day. Including data from new particle formation and studying the patterns in daily upslope
and downslope winds may provide additional clarity.

One of the primary drawbacks of these data is the gaps in data and possible remaining instrumentation errors. While the
data have been post-corrected, the POPS are not as accurate as more expensive, advanced particle counters. However, the price
point and still relative accuracy of the POPS made it a great option for a network of sites in remote locations. The gaps in data
made it impossible to compute a daily representation error from all six sites. This could affect the results of the representation
and network analysis since the daily representation error was computed from the sites that did have data each day. However, we
believe these possible errors do not affect the overall seasonal trends and the relationship between concentrations and elevations
that we observed.

This initial analysis supports the claim that aerosol concentrations are more variable both spatially and temporally in regions
of complex terrain than in flat land. However, the similar trends in the data from daily averages in Fig. 4 do indicate that
there is consistency across the region on a daily or larger timescale. This suggests that depending on the desired accuracy of
modeling efforts in the region, it may be necessary to take this variability into account. Furthermore, the change in variability
across seasons suggests that models would not retain the same accuracy over time. These data provide valuable insight into the
variability of aerosol in mountainous terrain and serve as a blueprint for future measurement networks in similar regions.



*Code and data availability.* The datasets used in this analysis are available on Zenodo: https://doi.org/10.5281/zenodo.12747225. These datasets as well as the raw data from the POPS available on the ARM Data Discovery: https://doi.org/10.5439/2203692.

The code to perform all analysis and generate figures is located in a GitHub repository: https://zenodo.org/doi/10.5281/zenodo.11238718.

*Author contributions.* LG performed the data curation and formal analysis and wrote the manuscript. EL was the PI, led project administra-
tion, assisted with site setup and monthly visits, supervised data analysis, and provided writing review and editing support. EE helped with site identification and setup and performed monthly site visits. NG designed and set up sampling site infrastructure and assisted with monthly site visits. AH was the Co-PI, assisted with site setup and monthly site visits, and provided project supervision and management. GM was the Co-I, assisted with site setup and monthly site visits, and advised on project management and data analysis. KP assisted with site setup and monthly site visits. BR assisted with site setup and monthly site visits and advised on data analysis TR designed and built instrument
enclosures. BS built CloudPucks and assisted with monthly site visits.

*Competing interests.* There are no competing interests.

*Acknowledgements.* We thank the US Department of Energy (DOE) Atmospheric System Research (ASR) program for funding through project DE-SC0022008. We thank Lawrence Berkeley National Laboratory (LBNL) for in-kind support. In-kind assistance from the LBNL was supported by the DOE Office of Science, Office of Biological and Environmental Research and Environmental System Science under
DOE contract DE-AC02-05CH11231. We thank the Rocky Mountain Biological Laboratory (RMBL) for the use of their land for our Gothic site.



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
