# Peer review of "Measurement Report: An investigation of the spatiotemporal variability of aerosol in the mountainous terrain of the Upper Colorado River Basin from SAIL-Net"

_EGUsphere, 2024_

## Author Response (AR1)

We would like to thank both reviewers for their time and valuable feedback provided. Below, we respond to both reviewers and note all changes to the manuscript.

Reviewer Comment
*Author Response*
Revised Text

**Reviewer 1**

In this manuscript the authors present results from an examination of spatiotemporal variability in aerosol measurements made across the East River Watershed in Colorado, a region of topographical heterogeneity. The methodology and comparisons are interesting, and offer some insights into the impacts of elevation and spatial distance on similarities between surface aerosol measurements. I find the presentation and analysis overall fairly straightforward and clear, though I do have a few concerns and recommendations for improvement.

*Thank you so much for your detailed review and suggestions. We will address your comments and concerns below and have revised the manuscript based on your suggestions. The updates are outlined in our responses.*

Figure 1: Considering the importance of elevation, I was surprised that this was not an elevation/contour map. I'm also not clear on the meaning of the many green shapes that dominate most of the map space. Unless those shapes are providing some sort of value that I am missing, I think showing the stations on a simple topographic map would be far more helpful here.

*The triangles indicate mountain peaks and the green polygons mark the boundaries of different land use areas. However, your suggestion to use a contour map to highlight the topography of the region is great. We updated the plot in the manuscripts so that the sites are now overlaid on a topographic map. We additionally added a figure of the state of Colorado, with the region where SAIL-Net occurred marked with a star, based on a suggestion from reviewer 2.*

Figure 6: Since the interannual variability is shown elsewhere (Figs 4 and 5), I would be far more interested in seeing a comparison of diurnal patterns between sites, rather than between years. It may also be helpful to group months by season, considering their similarities, which could allow for more detail. Separating sites by row would also allow variability to be shown by shading particular quantiles. All of these would greatly magnify the information provided here, which currently offers too many redundancies from previous figures (specifically overall monthly and interannual behavior).

*Thank you for this suggestion. We have redone Figure 6 (which is now Figure 8, due to some rearranging). This plot now displays seasonal diurnal cycles for individual sites with a shaded region plotting the interquartile range of data for the season.*

*To accompany this updated Figure, the paragraphs discussing it have also changed:*

"The diurnal cycles in aerosol concentrations changed seasonally and varied between sites. Figure 8 plots the average diurnal cycle of 170 nm to 3.4 µm sized aerosol concentrations for each SAIL-Net site seasonally. Concentrations were averaged hourly and then grouped by meteorological season. The shaded region around each line displays the interquartile range of the seasonal data. For this analysis, we removed data from 13 June 2022 to 16 June 2022 so that the abnormally high concentrations caused by wildfire smoke would not affect the trend.

The diurnal cycles were most pronounced in the summer and fall when there were higher total aerosol concentrations. In contrast, there were minimal to no diurnal cycles observed in the winter and spring. The lack of diurnal cycles in the winter months could partially be attributed to less vertical mixing of the boundary layer throughout the day (Gallagher et al., 2011). Irwin does seem to have some patternicity in the winter and spring, with concentrations increasing in the afternoon, but we believe this increase was due to consistent snowcat and snowmobile activity around the site during these seasons.

While the diurnal cycles look different across the SAIL-Net sites, there was an underlying consistency in the daily trends in the summer and fall. Aerosol concentrations tended to increase overnight and into the morning and peaked in the early afternoon. Concentrations then decreased throughout the late afternoon and evening. This behavior was especially clear for Pumphouse and Gothic in the summer, possibly due to the influence of anthropogenic activities around the sites, or due to unique conditions in the East River Valley where both sites were located. These observations were partially consistent with the diurnal analysis from Gallagher et al. (2011) at Whistler Mountain, which studied the seasonal and diurnal patterns of CCN. They found that diurnal cycles were more distinct in warmer months and less so in the winter. They also observed increasing CCN concentrations from 08:00 until approximately 16:00 LST as a result of new particle formation (NPF). While this daytime increase was also observed in the SAIL-Net data, we were unable to determine if NPF was driving this increase since the POPS cannot measure small enough particles to observe this. Likely, the height of the convective boundary layer coupled with anthropogenic activities in the nearby town of Crested Butte was driving the nighttime to midday increases in aerosol concentrations. However, more analysis would be necessary to be certain."

*This overall section was also rearranged to better flow between seasonal to a seasonal/diurnal analysis. The paragraphs that discuss the monthly average number size distributions now come before the diurnal analysis.*

Spatial distance: On line 248: "The average pairwise percent difference in aerosol concentration as a function of geographic distance between the two sites all yielded a

negative correlation on average." Does this mean that sites were *more* similar when *further* apart? This is confusing and nonintuitive, and deserves more explanation.

*Yes, we observed that the average pairwise percent difference between sites tended to decrease with distance between sites. You're right that this deserves further explanation. We edited Figure 10 to include plots of the percent difference as a function of the distance between the sites and discuss the results that we see. Essentially, this negative correlation may be an artifact of site placement, as we now elaborate on in the manuscript:*

"We do not see a relationship of similarity for sites that are nearby one another. All Pearson correlation coefficients are negative when comparing percent difference as a function of difference between sites. This result indicates that the common assumption that spatially close data are more similar does not apply here, which is particularly surprising. However, this observed negative correlation may be an artifact of site placement. The SAIL-Net sites that were within 5 km of one another also differed approximately 300 m to 700 m in elevation and were thus more different. For the majority of SAIL-Net sites that were greater than 5 km apart, their elevations were typically within 350 m of one another, so concentrations were more similar. Thus, the positive relationship between measurement similarity and elevation may have negatively influenced the relationship between spatially proximal sites."

Figure 12: This figure is very difficult to digest, in part due to the overplotting of the timeseries and the poorly differentiated sequential color scheme. Individual sites appear and disappear in a confusing manner, moving in front of or behind other colors seasonally. I highly recommend considering monthly (or seasonal) box plots rather than timeseries here to allow site differences to stand out without being overplotted. Choosing a better categorical color scheme with 6 distinct colors rather than a sequential viridis scheme will also greatly help, as some of the site colors are currently difficult to distinguish.

*Thanks for your feedback on this plot! In an attempt to have a plot that could be easily viewed by those with color vision deficiencies, I think this plot ended up needing to be more interpretable for everyone. We have opted to remove this Figure altogether and instead took your suggestion here and updated Figure 13, which is now Figure 12. This change is described in more detail in a different response below.*

On a related note, I'm not sure it's particularly useful to focus on the seasonality of normalized representation error, as it overemphasizes differences when overall concentrations are very low. Based on Fig 4, it seems likely that poor normalized performance in the winter is very likely an artifact of low aerosol counts in those months, magnifying normalized differences. I would personally be more interested in absolute errors, and at the very least the caption/discussion should provide better context for the apparently (but perhaps not actually) "worsened" winter performance.

*The worsened representation in the winter is likely at least partially an artifact of the low aerosol concentrations. We also see this artifact pop up in Figure 11 when plotting the coefficient of variation. Based on your suggestion here, we updated Figure 11 to also plot the monthly average range of concentrations between the sites. We believe that adding this plot provides a summary of the absolute differences between sites over different seasons, begins to address the impact of low aerosol concentrations on the analysis, and provides a valuable figure to refer back to when exploring the representation error as well.*

*We added the following discussion in section 3.2 regarding the impact of these lower concentrations affecting the CV:*

"The bottom plot of Fig. 11 displays the monthly average range of concentrations between the sites, which is typically lower in the winter and higher in the summer. There appears to be an inverse relationship between the monthly averaged CV and the monthly averaged ranges, indicating that despite the min-max scaling applied to the data, the number counts of aerosol in different seasons, affected the computed CV. Thus, although there appears to be higher variability in the colder months, this may predominantly be an artifact of the low wintertime concentrations."

*And we again mention this in section 3.3:*

"In general, the representation error appeared higher in the winter and lower in the summer. However, the lower aerosol concentrations across the sites in the winter likely impacted the representation error, so caution must be used when comparing the errors across seasons."

Figure 13: I like the direction here in seeing whether any particular site represents the whole better than the others, but I am confused by some choices. First, considering the strong seasonality we see in the overall record, it seems very important to evaluate site representation seasonally. There's a lot of empty white space in this figure, so I think this should be possible to add this dimension with a rethinking of panels and colors. Second, again I question the use of normalized metrics in this comparison, as it may be overemphasizing relatively tiny differences during periods of very low average concentrations. Please think this over, and either justify your decision or choose a metric without this problem.

*We agree that breaking the analysis down by season makes more sense, and we updated Figure 13 (which is now Figure 12) accordingly. Breaking this down by season should mitigate any misrepresentation that can occur seasonally, so long as representation errors are compared against others in the same season.*

*The choice of using the normalized representation error was to mimic the approach of Asher et al. (2021) in the analysis of POPSnet in order to compare the variability of SAIL-Net in complex terrain against the variability observed in POPSnet over the Southern Great Plains.*

The use of triangles and lines for means and ranges seems a little limited. Why not use box plots?

*In Figure 12 (previously Figure 13) we now use box plots which display the median, interquartile range, and 5th to 95th percentiles as whiskers.*

I don't see any value added in Table 2, and recommend it be removed or replaced. A simple ranking does not provide crucial details such as how much of a difference was observed between sites. A figure that showed absolute differences over time by site would do a better job of putting relative site performance in context.

*We agree and have removed this table. We believe that the updated Figure 12, which groups the representation error seasonally, provides better detail. We added a corresponding discussion for the updated figure as well:*

"...we compared the representation errors across the sites within each season to determine the most representative site for each season and size range. The most representative site should have a median close to zero and a small range. In Fig. 12, the whiskers of the box plot bound the 5th and 95th percentiles. To determine the most representative site, we assigned a score to each site and size range by summing the median's absolute value and the data range between the 5th and 95th percentiles. Using this approach, the most representative sites for each size range were

– 170 nm - 3.4 μm: Pumphouse (Spring), Irwin (Summer), Gothic (Fall), Gothic (Winter),

– 170 nm - 300 nm: Pumphouse (Spring), Irwin (Summer), Gothic (Fall), Gothic (Winter),

– 300 nm - 3.4 μm: Pumphouse (Spring), Snodgrass (Summer), Pumphouse (Fall), CBTop (Winter).

The most representative site was inconsistent over different seasons and between the two size ranges. This suggests that the aerosol properties of the region are complex and vary across seasons and sizes and thus there is not one consistent most representative site for the region."

On a related note, while the separate analysis of the three size ranges is interesting, I don't understand the aggregation strategy. Instead of summing the range and absolute value of the averages (which doubly penalizes sites that have a representative number of total particles, but a shifted size distribution), why not simply drop the bins and compare total aerosol concentrations to get an overall performance score? The current strategy seems to create an unjustified cost for differences in sizes that may or may not be important.

*The aggregation approach intended to give these three size ranges equal weight when looking at the representativeness of sites. Since there are typically so few supermicron sized particles to begin with, differences at sites due to larger particles would be outweighed due to the number of smaller particles. However, looking at the total aerosol concentration is also a valid approach that would provide a different holistic view of representativeness. To*

*address this comment, we now compute the representation error for three size ranges: the full 170 nm - 3.4 um, a smaller size range of 170 nm - 300 nm, and grouped all larger particles into a 300 nm - 3.4 um size range for analysis. We still sum the median and the range because we want to evaluate each site by how close it typically is to the network mean and by how far it can drift from the network mean. This change is reflected in Figure 12 and the analysis, which was also copied in the above response.*

Significance of complex terrain: line 380 states that "This initial analysis supports the claim that aerosol concentrations are more variable both spatially and temporally in regions of complex terrain than in flat land", but I'm not clear on where this is explicitly supported. Is there a similar study performed on flat land that these results are being compared to?

*This line is corroborating the findings of citations in line 21. The most similar study to SAIL-Net over flat land was POPS-Net (Asher et al., 2021). We added a more explicit comparison of the two in Section 3.3:*

"One of the observations driving the deployment of SAIL-Net was that aerosol complexity is increased in mountainous terrain compared to flat land (Zieger et al., 2012; Yuan et al., 2020; Nakata et al., 2021). The results of SAIL-Net further support this conclusion. In comparing the range of representation errors against the results of POPSnet (Asher et al., 2022), which collected data between October and March, SAIL-Net observed the same or higher errors across many of the sites in both the winter and spring. This suggests that aerosol complexity is increased in mountainous terrain since SAIL-Net sites were more spatially dense than POPSnet but still observed equal or greater error in the same season".

**Reviewer 2**

This manuscript describes the deployment of 6 low cost optical particle counters (POPS) in the East River Watershed in Colorado. The POPS measures particle counts and size distributions from ~140 nm to 3.4 microns. The analyses compare the measurements across the network for different particle size ranges and seasons. Overall the manuscript is topically relevant to ACP and all of the data and code are publicly available.

*Thank you for your comments and suggestions! We will address your comments and concerns below and have revised the manuscript based on your feedback. These changes are described below.*

I have a few overarching comments:

(1) The article needs a thorough review for copy editing and quality assurance. For example, the paragraph from lines 303-309 is repeated as the following paragraph. Other examples are listed below.

Line 52 - CNN instead of CCN.

Line 149 CloudPuck is misspelled.

*Thank you for catching these. We truly thought we caught everything, but having a fresh set of eyes does wonders. We have corrected these errors and checked for others.*

(2) In general the figures could benefit from better labeling, captions, and improving the axis ranges and placement of tick marks. For example, in Fig 12 the y-axis is zoomed out pretty far and there are very few tick marks, so it is hard to interpret the magnitude of the errors.

*A number of the figures and captions have been updated based on suggestions from Reviewer 1, and care has been taken to improve the clarity of the figures and captions. Below is a list of figures that have been updated in some way with a brief description of how:*

- *Figure 1: caption and figure updated, figure displays plot of Colorado with SAIL-Net region marked, and topographic map of sites*
- *Figure 3: cosmetic updates to colors*
- *Figure 4: cosmetic changes for ease of reading*
- *Figure 6 (now Figure 8): figure and caption updated, figure displays diurnal cycle for each site, grouped seasonally instead of monthly*
- *Figure 10: updated to also show plot of percent difference vs distance*
- *Figure 11: updates to figure and caption, figure shows monthly average CV and range of concentrations monthly*

- *Figure 12: updates to figure and caption, figure combines ideas from the previous Fig 12 and 13, now displays box plots of representation error broken down seasonally*
- *Figure 13 (previously Figure 14): updates to figure and caption, fig now shows two examples of TBS flights and caption was edited for clarity*
- *Figure 14 (previously Figure 15): figure and caption updated, figure now displays box plots of absolute difference for each day with median percent error plotted as a timeseries*

Specific comments:

Line 90 - can you be a bit more specific about what makes the POPS "research grade"? How is research grade quantitatively different than something that is not research grade, in this instance?

*This is a great question. "Research grade" is vague and we have opted to remove that phrase in the manuscript to avoid ambiguity. The intent of using this phrase was to differentiate the POPS from other lower-cost sensors like PurpleAir. We instead rephrased this sentence as the following:*

"In the last few years, it has been recognized for its accuracy and reliability as a low-cost sensor, and used in a number of field deployments and campaigns (Mei et al., 2020; Brus et al., 2021; Asher et al., 2022; Todt et al., 2023)."

Fig 1 is extremely zoomed in. Could you add an inset showing the location of the study area within Colorado? What are the green lines, black lines, and orange triangles?

*The triangles indicate mountain peaks and the green and black lines mark land use boundaries. We've updated this plot to overlay on a topographic map, which we believe makes more sense given SAIL-Net's focus on studying aerosol in complex, mountainous terrain. We also added a map of the state of Colorado which marks where the SAIL-Net study region was within the state.*

Section 2.2 - Is the direction of the drift expected to be monotonic? Or will it fluctuate over time?

*Thank you for the clarifying question. The drift is expected to be monotonic. We clarified that the drift of the POPS is monotonic due to decreased light intensity measured by the digitizer in the following sentence:*

"In either case, the lower intensity of light causes particles to be sized smaller than their true size, and thus the drift of the POPS is monotonically decreasing over time."

Line 120 - when discussing drift correction, it would be helpful to give the readers a sense of what the data look like. I assume it's some sort of size distribution or number of counts per size bin, but that is not specified.

*We added a few sentences explaining the POPS data in greater detail, which are binned into one of 16 bins with raw data reported as number counts (and were converted to number concentration in the publicly available datasets). The addition reads:*

"The data collected by the POPS during SAIL-Net were binned into one of 16 bins as number counts based on the measured size of the particle. These number counts were converted to number concentrations in publically available datasets (Gibson and Levin, 2023). In diameter space, the widths of the bins are not equal but increase non-monotonically with size. The size range of particles for smaller bins is approximately 15 nm, while the size range for larger bins is approximately 600 nm. The following description will provide insight into why the bins are unequal widths in diameter-space and why this increase is not strictly monotonic."

Line 130 - how wide are the bins around 500 nm? This would help readers understand how likely it is that a particle in that size range might be binned incorrectly.

*We elaborated on the sizing of bins near 500 nm by adding the following two sentences:*

"The the bin that contains 500 nn sized particles has a lower bound of 497 nm. Thus the drift in a POPS is caught early on because 500 nm sized particles will very quickly be sized into the bin below as the drift starts to occur."

Line 130-135 - it would help to show this correction graphically

*The purpose of lines 130-135 is to provide justification that the post correction is indeed as simple as shifting bins (i.e. all particles in bin 1 are put into bin 2 and adopt bin 2's boundaries, etc.), as opposed to having to derive new bin boundaries. Given the abstractness of this justification, we believe that attempting to graphically show it would likely not improve understanding, and is also outside the focus of the overall manuscript. However, we did add a written example of this shifting process to provide a more concrete explanation of the result of the post correction:*

"As an example, if the PSL check sized 500 nm particles into one bin smaller, then the post correction process would move all particles from bin one into bin two, bin two into bin three, and so on. Then the particles that were previously sized as 143 nm to 155 nm in bin one would now be sized as 155 nm to 170 nm, adopting the size range of bin two. Because of this upward shifting, the lowest size that the POPS measured increased throughout the deployment."

Line 143 says that the lower size limit was 170 nm. However, above it stated 3 microns (line 92).

*Line 92 describes the size range that the POPS can measure; as low as 140 nm and as large as 3.4 um. Lines 140-142 explain that because of the post-correction of the data, the lowest size particle that the POPS measured increased throughout the campaign. Thus the majority of the analysis uses a slightly larger minimum size of 170 nm because of this shift in the data due to post correction.*

Line 199 notes that particle concentrations increase around 6 pm in most months, and attributes this to growth of particles into the POPS size range. However couldn't this also be due to the increase in concentration associated with the formation of the nighttime boundary layer?

*Evening and nighttime increases could be associated with changes in the boundary layer. We have updated the discussion around this section based on feedback from Reviewer 1, and within this have noted possible influence by the boundary layer. However, we also believe that more analysis would be necessary here.*

"It is likely that the height of the convective boundary layer coupled with anthropogenic activities in the nearby town of Crested Butte were driving the nighttime to midday increases in aerosol concentrations. However, more analysis would be necessary to be certain."

Figure 7 - does this plot show dN/dlog(Dp)? Or N vs Dp?

*Fig 7 (which is now Fig 6 due to some reorganization of this section) plots N vs Dp. We clarified this in the sentence,* "Figure 7 displays the monthly average number size distribution (N vs Dp) of aerosol overlaid by month." *This was also clarified in the caption for Figure 7:* "The number size distribution of the network mean is averaged monthly. When multiple years of data are present, both are plotted. These plots used the full 140 nm to 3.4 μm size range of the POPS."

Line 248-249: "The average pairwise percent difference in aerosol concentration as a function of geographic distance between the two sites all yielded a negative correlation on average." Is this shown anywhere?

*We edited Figure 10 to include plots of the percent difference as a function of the distance between the sites and discuss the results that we see. Essentially, this negative correlation may be an artifact of site placement, as we now elaborated on in the manuscript:*

"We do not see a relationship of similarity for sites that are near one another. All Pearson correlation coefficients are negative when comparing percent difference as a function of difference between sites. This result indicates that the common assumption that spatially close data are more similar does not apply here, which is particularly surprising. However, this observed negative correlation may be an artifact of site placement. The SAIL-Net sites that were within 5 km of one another also differed approximately 300 m to 700 m in elevation. For the majority of SAIL-Net sites that were greater than 5 km apart, their elevations were typically within 350 m of one another. Thus, the positive relationship between measurement similarity and elevation may have negatively influenced the relationship between spatially proximal sites."

I'm a little unclear by what is plotted in Fig 11. Is this the CV across/between sites? Or does it mix the variability both across sites and within each site in a given day? Please clarify.

*Fig 11 plots the CV across sites. Each set consists of the daily average aerosol concentration at each site and the CV is computed from this. We clarified this in the manuscript by updating this sentence:*

"Using the daily average concentration of 170 nm to 3.4 µm sized particles at each site, the data were grouped by time, so that each time step provided a set of data across the sites for which to compute the CV."

Section 3.3 on representation error. It seems like this approach to computing representation error assumes that the network average is indeed a true representation of "truth." Also, it seems like this metric for representation error could be challenging to use when the denominator is small, as is often the case with this dataset. Is the representation error calculation impacted by the fact that the number and identity of sites changes with time?

*The representation error assumes that the network mean is a proxy for the true value of the region. We made this more explicit in the manuscript:*

"The representation error treats the network mean as a proxy for the true regional value and then quantifies how different a single site is from the network mean."

*As with the computation of the network mean, the representation error is also computed whenever possible, meaning that the number of sites is not consistent over time. This could have some impact, but we believe that there is sufficient data for the impact to be minimal. We also now explicitly note this in the manuscript:*

"The representation error was then computed for each site on every day when there was valid data. As with the computation of the network mean, not all days had data for all six sites, but in order to maximize the temporal span of data, we computed the representation error for sites whenever possible. Since the number of sites and number of days of data were not consistent across sites, this could have some effect on the results of the following analysis. However, we believe that given the approximately 600 sampling days, there was sufficient data that these missing values should not have a massive impact on the overall results."

*Lastly, indeed the representation error could appear larger when the network mean is small. At the same time, the difference between the site measurement and the network mean (the numerator) should also be smaller during these times, but you are correct that this normalized approach could have that impact.*

Fig 12 - the figure is hard to interpret. The y-axis would benefit from having more tick marks.

*Based on feedback from reviewer 1, Figure 12 has been entirely changed and instead is an improved version of Figure 13, broken down by season and containing data on various quartiles of the data. There are also more ticks on the y-axis in this plot.*

Figure 14 - is this a fair comparison? Is the relevant height the height above sea level, or the local height above the ground (or some mix of the two)?

*The comparison between SAIL-Net and the TBS flights aims for as close to an apples-to-apples comparison as possible. The relevant height is the height above sea level. We recognize that one measurement is ground-based (SAIL-Net) and the other is airborne (balloon) when using the height above sea level, but this choice was made to learn if the SAIL-Net sites could represent the vertical column of air in the region. The section describing this comparison has been revised in the manuscript for clarity:*

"During every TBS flight, the balloon was sent up vertically through the atmosphere. The balloon remained approximately in the same geographic location so that each flight generated a profile of the vertical air column in the region, where each measurement was associated with an altitude above sea level $a\_f$ and a time $t\_f$. To compare the data from the TBS flight with SAIL-Net, we built a pseudo "vertical column" from the SAIL-Net sites. We did this by associating the altitude above sea level of each site, $a\_s$ with its measured total aerosol concentration at a time $t\_s$, ignoring the geographical

location of the site. Time t_s was determined by the time at which the altitude (above sea level) of the TBS balloon passed within 2.5 m of the altitude above sea level of the site. Mathematically, t_s was the time at which a_s - 2.5 < a_f < a_s + 2.5. We then averaged the concentrations at site s in the one minute window around t_s, and set this to be the value of the pseudo vertical column at the altitude, time pair (a_s, t_s). The error between the concentration reported by the POPS on the TBS flight and the concentration at the SAIL-Net site was computed for each (a_s, t_s) in the SAIL-Net vertical column.

We recognize the "vertical column" generated by the SAIL-Net sites is a crude approximation of a vertical column since it does not account for the differing geographic locations of the sites, and the measurements from the sites are ground-based instead of airborne. However, this approach provided a straightforward method for comparison between spatially dispersed ground-based measurements and airborne measurements."

I'm not sure what to make of Fig 15. The errors seem to be generally very large. On a few days there is more variability in the errors, but in general they seem to clump together on a given day.

*We have significantly updated Figure 15 (now Figure 14 in the manuscript) so that it is easier to interpret. Instead of displaying only means and medians, which clumped the data together, we plotted boxplots of the absolute differences between the sites and a timeseries of the median percent difference for each day. We hope this will make it easier to understand this figure.*

I think Figs 14 and 15 are supposed to help bolster the argument that there is not a single "representative" site for this network. However, I think that point is adequately made with Figs 12 and 13, and that Figs 14 and 15 don't add much.

*The intent of the comparison between the SAIL-Net sites and the TBS flights was more exploratory research than anything else. We were curious to learn if our ground-based measurements could at all be representative of the vertical column of air in the region. We revised the manuscript to make this more clear by adding a small introduction at the start of Section 3.3, and by also breaking this section into two subsections for the regional representation analysis of the network mean and the vertical representation analysis.*

*The introduction to Section 3.3 now reads:*

"The previous subsections highlighted the temporal and spatial variability in the ERW. We now use these data to investigate the optimal network design in the region and determine if a single site can accurately represent the aerosol properties of the region. This section is broken into two separate

analyses. The first investigates the spatial representativeness of the sites, using a similar analysis approach to that of Asher et al. (2022) during POPSnet. The second analysis is more exploratory, and utilizes the varying altitudes of the SAIL-Net sites to compare our ground-based measurements to airborne measurements from tethered balloon flights, which characterized the vertical column of air in the region."

---

## Author Response (AR2)

*We would like to thank both reviewers for their additional reviews and feedback. Below, we note the additional minor changes to the manuscript according to feedback from the reviewers and editor.*

Reviewer/Editor Comment
*Author Response*
Revised Text

Thank you for submitting your revised manuscripts based on the reviewers' comments. Both reviewers are mostly satisfied with your revisions. However, there's one remaining item related to the extent of the drift in POP's sizing and the corrections that were made to the size bins. Please add a figure to SI that demonstrates the extent of this correction for the different sites and discuss this range in the main text (Section 2.2).

*Thank you for this additional feedback. We have added additional information to both the manuscript and the Supplement to address this item. In the manuscript, we first corrected the PSL size from 500 nm to 495 nm (the exact size used). This change does not affect the overall understanding of the post-correction process. Second, we added a few additional sentences to reference the additions to the SI and to add more clarity to the post-correction. The sentences are written below, but will likely make more sense if read in the entire context of Section 2.2.*

"Figure 1 of the Supplement shows an example of how these PSL checks look in the raw data."

"The bin that contains the 459 nm sized particles has a lower bound of 350 nm. Thus, the drift is recognized when PSL is sized smaller than 350 nm, shifting the bin in which the PSL signal occurs."

" For the majority of the following analysis, the minimum particle size used will be 170 nm (minimum of the third POPS bin) instead of the 140 nm that is standard with the POPS to account for this shift. This shift allows for a fair comparison across sites, even with a drifted POPS. If a POPS shifted more than two bins, those data were not used in the following analysis. Pumphouse experienced the most drift, where in the last month of the deployment, particles were sized four bins lower. Snodgrass experienced the least drift, with the PSL check still sized properly at the end of the deployment. Table 1 of the Supplement contains a table of the monthly documented drift of each POPS."

*The SI now contains a figure of what the raw data look like during a PSL check where the POPS has drifted. It also contains a table that documents in which bin the PSL signal occurred for all POPS during each month that they were deployed. This allows the reader to explicitly see when each POPS experienced drift and to what extent, and also understand in which months the data were post-corrected.*

*As a final note, if there is a specific template for the SI, please let me know. I could not find one online and I know everyone if out of the office until January 2, after which these revisions are due, so I added the SI in a basic pdf, but am happy to change the formatting if needed.*

---

## Author Response (AR3)

Reviewer Comment
*Author Response*
Revised Text

Thanks for submitting the revisions. I'm happy to accept your paper.
As for the SI.... please add "Supplement of" followed by the title and authors' list to the first page of the pdf.

*We are thrilled that our manuscript has been accepted. We added "Supplement of", the title, and the authors' list to the SI.*